# Mitochondrial Ultrastructure and Activity Are Differentially Regulated by Glycolysis-, Krebs Cycle-, and Microbiota-Derived Metabolites in Monocytes

**DOI:** 10.3390/biology11081132

**Published:** 2022-07-28

**Authors:** C. Angélica Pérez-Hernández, M. Maximina Bertha Moreno-Altamirano, Edgar O. López-Villegas, Egle Butkeviciute, Mohammad Ali, Barbara Kronsteiner, Susanna J. Dunachie, Hazel M. Dockrell, Steven G. Smith, F. Javier Sánchez-García

**Affiliations:** 1Laboratorio de Inmunorregulación, Departamento de Inmunología, Escuela Nacional de Ciencias Biológicas, Instituto Politécnico Nacional, Mexico City 11340, Mexico; biol.angelica.1221@gmail.com (C.A.P.-H.); mmorenoal@ipn.mx (M.M.B.M.-A.); 2Unidad de Microscopía, Escuela Nacional de Ciencias Biológicas, Instituto Politécnico Nacional, Mexico City 11340, Mexico; lvoliver@hotmail.com; 3Department of Infection Biology, London School of Hygiene and Tropical Medicine, London WC1E 7HT, UK; egle.butkeviciute@lshtm.ac.uk (E.B.); hazel.dockrell@lshtm.ac.uk (H.M.D.); 4Peter Medawar Building for Pathogen Research, Nuffield Department of Clinical Medicine, University of Oxford, Oxford OX1 3SY, UK; mohammad.ali@wolfson.ox.ac.uk (M.A.); barbara.kronsteiner-dobramysl@ndm.ox.ac.uk (B.K.); susie.dunachie@ndm.ox.ac.uk (S.J.D.); 5Oxford Centre for Global Health Research, Nuffield Department of Clinical Medicine, University of Oxford, Oxford OX3 7LG, UK; 6Division of Biosciences, Brunel University London, London UB8 3PH, UK; steven.smith@brunel.ac.uk

**Keywords:** innate immunity, mitochondria, glycolysis, Krebs cycle, short-chain fatty acids, mitochondrial reprogramming

## Abstract

**Simple Summary:**

Beyond their metabolic role, several metabolites are also signalling molecules, contributing to cell–cell communication. This work analyses how monocytes mitochondria differentially respond to metabolites.

**Abstract:**

Several intermediate metabolites harbour cell-signalling properties, thus, it is likely that specific metabolites enable the communication between neighbouring cells, as well as between host cells with the microbiota, pathogens, and tumour cells. Mitochondria, a source of intermediate metabolites, participate in a wide array of biological processes beyond that of ATP production, such as intracellular calcium homeostasis, cell signalling, apoptosis, regulation of immune responses, and host cell–microbiota crosstalk. In this regard, mitochondria’s plasticity allows them to adapt their bioenergetics status to intra- and extra-cellular cues, and the mechanisms driving such plasticity are currently a matter of intensive research. Here, we addressed whether mitochondrial ultrastructure and activity are differentially shaped when human monocytes are exposed to an exogenous source of lactate (derived from glycolysis), succinate, and fumarate (Krebs cycle metabolic intermediates), or butyrate and acetate (short-chain fatty acids produced by intestinal microbiota). It has previously been shown that fumarate induces mitochondrial fusion, increases the mitochondrial membrane potential (Δψ_m_), and reshapes the mitochondrial cristae ultrastructure. Here, we provide evidence that, in contrast to fumarate, lactate, succinate, and butyrate induce mitochondrial fission, while acetate induces mitochondrial swelling. These traits, along with mitochondrial calcium influx kinetics and glycolytic vs. mitochondrial ATP-production rates, suggest that these metabolites differentially shape mitochondrial function, paving the way for the understanding of metabolite-induced metabolic reprogramming of monocytes and its possible use for immune-response intervention.

## 1. Introduction

Monocytes/macrophages are key components of innate immunity and inflammation, and, beyond their phagocytic role, these cells also contribute to host homeostasis. This versatility is achieved at least in part by metabolic reprogramming [1,2,3]. IFN-γ/LPS-activated macrophages (M1 macrophages), regarded as pro-inflammatory cells, are involved in pathogen clearance and are characterized by the up-regulation of glycolysis, whereas IL-4-activated macrophages (M2 macrophages), which have a role in anti-helminthic response and tissue repair, are mostly dependent on oxidative phosphorylation [4].

Metabolic reprogramming not only sustains the energy demands of activated macrophages, it also leads to the production of intermediate metabolites that harbour cell-signalling properties and participate in the regulation of immune responses. As such, M1 macrophages accumulate succinate, through the inhibition of succinate dehydrogenase (SDH), which triggers HIF1-α stabilization-dependent inflammatory signalling [5]. Moreover, fumarate, the product of succinate oxidation, accumulates in the course of innate immune training, contributing to the epigenetic reprogramming of trained monocytes [6]. Both succinate and fumarate can be exported to the extracellular milieu, where they can be recognized by cell-membrane receptors such as GRP91 and GRP109, respectively, leading to intracellular signalling [7,8,9,10,11,12]. 

The glycolytic pathway also contributes signalling molecules, such as lactate, which accumulates in the extracellular milieu following the increase in the glycolytic rate associated with chronic inflammation and the tumour microenvironment [13,14]. Lactate is recognized by the GRP81 cell receptor [15] and is transported in and out of the cell by different monocarboxylate transporters [MCTs) [16,17,18]. Lactate elicits the polarization of macrophages towards the M2 phenotype, contributing to the immunosuppressive microenvironment of tumours [14,15,19,20].

Furthermore, the host-associated microbiota regulates immune responses through a dynamic crosstalk with immune cells, a process in which mitochondria seem to play a vital role [21,22]. Short-chain fatty acids (SCFAs) are amongst the main microbiota-derived components, of which butyrate and acetate are predominant. Both butyrate and acetate function as histone deacetylase inhibitors (HDCA), down-regulating NF-kB expression and preventing inflammatory cytokine production [23,24]. SCFAs are recognized by GRP41, GRP43, and GRP109 receptors [25,26]. 

In line with the view that mitochondria could serve as a hub in the integration of metabolic signals from different biological domains [22], and considering that mitochondria constantly undergo ultrastructural changes that contribute to cell functional plasticity [27,28,29,30], we analysed mitochondrial ultrastructure as well as some functional parameters in monocytes exposed to an extracellular source of lactate, succinate, fumarate, butyrate, and acetate. Results showed that each metabolite imprinted a distinctive mitochondrial signature, suggesting a mechanism by which mitochondria might integrate extracellular metabolic cues leading to the metabolic reprogramming of monocytes.

## 2. Materials and Methods

### 2.1. Cells

Peripheral blood mononuclear cells (PBMC) were obtained from buffy coats or from 20 to 30 mL of peripheral blood from healthy donors, after written informed consent was obtained and under the Declaration of Helsinki Guidelines (Ethics Committee approval numbers LSHTM-5520, LSHTM-14576, and 16/YH/0247). Healthy volunteers donating blood for research studies declared that they are not aware that they suffer from any persistent medical condition or infection that would affect their suitability to be a blood donor. At LSHTM, they were also asked if they have been BCG vaccinated, and if they have lived abroad or visited a malaria-endemic area or ever had malaria. Blood was collected in heparinised tubes, except for extracellular-flux-analysis experiments, for which blood was collected in EDTA. Peripheral blood was diluted with sterile PBS and layered over Ficoll-Paque (GE Healthcare, Chicago, IL, USA). After differential density centrifugation, PBMC were recovered and washed four times with PBS. PBMC were cultured overnight at 37 °C in a 5% CO_2_ atmosphere to allow monocyte adherence. Adherent cells (monocyte-enriched cells) were washed twice with PBS and maintained in RPMI-1640 medium supplemented with 0.3 g/L of L-glutamine (Thermo Fisher Scientific, Walthman, MA, USA), at 37 °C and a 5% CO_2_ atmosphere for the time of the experiments. For magnetic beads enrichment of monocytes, peripheral blood was layered over Lymphoprep (Stem Cell Technologies, Vancouver, BC, Canada) or Ficoll-Paque (GE Healthcare, Chicago, IL, USA), and PBMC was isolated using density-gradient centrifugation. PBMC were subjected to several low-speed washes in PBS in order to minimise platelet contamination and were subsequently subjected to monocyte enrichment by negative selection using a monocyte-enrichment cocktail and magnetic separation on an EasySep magnet or RoboSep instrument (Stem Cell Technologies, Vancouver, BC, Canada) or by positive selection using CD14 microbeads (Miltenyi Biotec, Bergisch Gladbach, Germany).

### 2.2. Metabolites

Metabolite stocks (1.0 M) were prepared as follows: mono-methyl fumarate (“fumarate”) (Sigma-Aldrich, St. Louis, MO, USA) was dissolved in DMSO and diethyl succinate (“succinate”) (Sigma-Aldrich) in ethanol. Sodium-L-lactate (“lactate”) (Sigma-Aldrich), sodium butyrate (“butyrate”) (Sigma-Aldrich), and sodium acetate (“acetate”) (Sigma Aldrich) were dissolved in PBS. Metabolite stocks were kept at −20 ℃ and later diluted 1:100 (100X, working solution) in RPMI-1640 culture medium, and all were used at a final concentration of 100 μM. In all the experiments, the cells were cultured in RPMI-1640 (supplemented with glucose and glutamine) being the only difference between treatments the addition of a single metabolite (100 μM). Cells cultured in RPMI-1640 without the addition of any external source of lactate, fumarate, etc., were used as a negative control, so that the results can be interpreted as a consequence of the addition of an exogenous source of each metabolite. The 100 μM concentration is around the physiological concentrations in different human fluids (Human Metabolome Database, http://hmdb.ca accessed on 30 June 2022). 

### 2.3. Cytoplasmic and Mitochondrial Calcium Influx Assays 

Human monocytes were independently loaded with the cytoplasmic calcium indicator Fluo-4/ AM (10 μM) (Thermo Fisher Scientific Inc) or with the mitochondrial calcium indicator Rhod-2/AM (10 μM) (Thermo Fisher Scientific Inc) for 30 min at room temperature; after washing with PBS, cells were suspended in RPMI-1640 medium supplemented with 2 mM CaCl_2_ and transferred to FACS tubes (Corning) (1 × 10^6^ cells in 0.5 mL). Basal cytoplasmic and mitochondrial calcium was recorded for 30 s; then, 100 μM of metabolites were added separately to each tube, and mitochondrial or cytoplasmic calcium influx was recorded for 180 s by flow cytometry (FACSCalibur; BD Biosciences, Franklin Lakes, NJ, USA). Raw data were analysed with FlowJo v10.6.1 (FACS Software; BD Biosciences) for mean fluorescence intensity (MFI) over time. The addition of 100 ng/ml of Ionomycin (Sigma-Aldrich) was used as a positive control of calcium influx. Data are represented as the fold change of MFI, where the numerical value of MFI for the base level is normalized to a value of 1.0.

### 2.4. Mitochondrial Ultrastructure Analysis 

Monocytes cultured in Petri dishes (Corning, NY, USA), were treated for 3 h with 100 μM of each metabolite, separately, or left untreated as a negative control. Cells were washed with Sorensen buffer and fixed for 30 min with 3% potassium permanganate in Sorensen buffer. After fixing, monocytes were carefully scraped off and centrifuged in a 15 mL tube (Corning) and then transferred into microtubes (Eppendorf, Hamburg, Germany). Cells were washed with Sorensen solution and dehydrated with gradually increasing concentrations of ethanol (from 10% to 100%). The cellular pellet was embedded in EPON 812 (Electron Microscopy Sciences, Hatfield, PA, USA) and cured in an oven at 60 °C for 24 h, after which 70 nm ultrathin sections were obtained using a microtome. Micrographs were obtained with a Jeol JEM1010 electron transmission microscope at 60 kV. Mitochondrial shape descriptors and size measurements were obtained using ImageJ (1.8.0_112) (National Institute of Heath, Bethesda, MD, USA), by manually tracing each mitochondria contour, as described by Picard et al., 2012 [31]. The mitochondria from 50 cells per condition were analysed. Surface area (mitochondrial size) is reported in μm^2^; perimeter in μm; aspect ratio (AR) is represented by [(major axis)/minor axis)] and reflects the length-to-width ratio; form factor (FF) [(perimeter^2^)/(surface area)] reflects the complexity and branching aspect of mitochondria; circularity [4 (surface area/perimeter^2^)] and roundness [4 (surface area)/(major axis^2^)] are two-dimensional indexes of sphericity with values of 1 indicating perfect spheroids; and Feret’s diameter represents the longest distance (mm) between any two points within a given mitochondrion [31]. Raw data were analysed by Prism 5 (Graph Pad Software). As described by Tobias et al., 2018 [32], the cristae density was calculated from the cristae number normalized to mitochondria surface area. To evaluate the cristae-alignment degree, the interception angles among cristae were measured using ImageJ. 

### 2.5. Mitochondrial Membrane Potential (Δψ_m_) Analysis 

Adherent PBMC were carefully scraped from the Petri dishes, counted, and transferred to FACS tubes (Corning) (1 × 10^6^ cells per tube), labelled with anti-human CD14-APC (HCD14) monoclonal antibody (BioLegend, San Diego, CA, USA) for 20 min at 4 °C, washed and incubated in the presence of 100 nM tetramethylrhodamine methyl ester (TMRM) (Thermo Fisher Scientific, Inc.) for 20 min at 37 °C. The Δψ_m_ was assessed by flow cytometry (FACSCalibur; BD Biosciences). Myeloid cells were first gated using forward-scatter and side-scatter parameters, and then the CD14^+^ cell population was gated using forward scatter vs. APC. CD14^+^ cells were analysed for TMRM mean fluorescence intensity (MFI), as indicative of mitochondrial membrane potential. Δψ_m_ was assessed prior to metabolite stimulation (time 0). Metabolites (100 μM) were separately added to the cells, and MFI was recorded at 1, 2, 3, and 4 h after metabolite exposure. Raw data were analysed with FlowJo v10.6.1 (FACS Software; Tree Star, Inc.). In each experiment a TMRM-loaded cell sample was used as an assay control. MFI was recorded at base level, then a respiratory chain uncoupler CCCP (Carbonyl cyanide 3-chlorophenylhydrazone (Sigma) was added, in order to confirm that the observed fluorescence was really due to Δψ_m_. A drop in fluorescence was indicative of mitochondrial depolarization. Only then was the rest of the experiment carried out. Data are represented as the fold change of MFI, where the numerical value of MFI at time 0 is normalized to a value of 1.0.

### 2.6. Extracellular Flux Analysis

Enriched monocytes (magnetic-based negative selection) at a median purity of 88% were suspended in RPMI-1640 supplemented with 10% FBS, 1 mM Pen/Strep, and 2 mM L-glutamine (Sigma Aldrich) and cultured at a density of 0.5 × 10^6^ cells/ml in a 6-well TC untreated plate at 2 mL per well. Cells were treated with 100 µM of either lactate, succinate, fumarate, acetate, and butyrate, or equal volumes of the respective vehicle controls (DMSO, ethanol, PBS). After 20 h of incubation, monocytes were harvested, washed, and resuspended in Seahorse assay medium comprised of unbuffered RPMI-1640 at pH 7.4 without phenol red and supplemented with 10 mM glucose, 2 mM L-glutamine, and 1 mM sodium pyruvate (all from Agilent Technologies, Santa Clara, CA, USA). Monocytes (100,000 cells/well) were then adhered to a poly-D-lysine coated XF96 cell culture microplate for an hour and sequentially treated with 2 µM oligomycin (port A) and 0.5 µM rotenone/antimycin A (port B) using the ATP-rate assay kit (Agilent Technologies) in accordance with the instructions of the manufacturer. Five to six technical replicates per condition were measured on an XF96 extracellular-flux analyser (Seahorse, Agilent Technologies) and mitochondrial (mito)-ATP and glycolytic (glyco)-ATP rates were calculated from extracellular-acidification rate (ECAR) and oxygen-consumption rate (OCR) readings using Wave software (Agilent Technologies). The addition of the ATP synthase inhibitor oligomycin inhibits mitochondrial ATP production thus reducing OCR and allowing the quantification of mitoATP production rate. ECAR levels measured in this assay combined with the buffer factor of the assay medium (determined by the manufacturer) were used to calculate total proton-efflux rate (PER). Since mitochondrial respiration also contributes to ECAR levels, complex I and III of the electron-transport chain were blocked by the addition of rotenone and antimycin A, respectively. This combined with the PER was then used to calculate the glycoATP production rate.

### 2.7. Statistical Analyses 

Calcium influx and Δψ_m_ data were analysed by Kruskal–Wallis and Dunn’s post hoc tests. Mitochondrial ultrastructure data were analysed by unifactorial ANOVA. All statistical analyses were performed using Graph Pad Prism software (GraphPad, La Jolla, CA, USA). Statistically significant differences were set at *p* < 0.05.

## 3. Results

### 3.1. Lactate-, Succinate-, Fumarate-, Butyrate-, and Acetate-Stimulation Triggers Cytoplasmic and Mitochondrial Calcium Influx

Cytoplasmic and mitochondrial calcium influx was tracked in real time for up to 180 s following metabolite exposure. All the metabolites tested triggered cytoplasmic (Figure 1A) as well as mitochondrial calcium (Figure 1B) influx within a few seconds. Fumarate appears to promote the highest calcium influx to the cytoplasm, whereas the mitochondrial calcium concentrations, following metabolite treatment, were similar for all the metabolites tested. 

### 3.2. Mitochondria from Lactate-, Succinate-, and Butyrate-Stimulated Monocytes Became Smaller and Less Complex Than Mitochondria from Fumarate-Stimulated Monocytes 

Monocytes were separately exposed to five different metabolites (lactate, succinate, fumarate, butyrate, and acetate) for 3 h or left untreated as a negative control. Lactate, succinate, and butyrate induced mitochondria to become smaller and more rounded compared with those in untreated cells (Figure 2A). This observation was confirmed by morphometric analyses, i.e., measurement of surface area and perimeter, which showed statistically significant reductions in mitochondrial size in the cells exposed to these metabolites (Figure 2B). In contrast, mitochondria from monocytes exposed to fumarate became more elongated, branched, and complex than the mitochondria from monocytes cultured in medium alone (Figure 2A), with statistically significant higher values of surface area and perimeter (Figure 2B). On the other hand, mitochondria from acetate-exposed monocytes increased their surface area and elongation without an increase in complexity (Figure 2B). 

### 3.3. Fumarate and Acetate Induce Opposite Effects on the Ultrastructure or Ganization of Mitochondrial Cristae 

To get insight into the inner organization of mitochondria, the cristae alignment and density within mitochondria were assessed. Fumarate treatment induced a reduction in the cristae-interception angle, which implies that mitochondria cristae were closer to each other (Figure 3A). However, no significant difference in mitochondrial-cristae density was observed in comparison with untreated monocytes (Figure 3B). As for acetate treatment, mitochondrial-cristae density decreased, whereas the interception angle became greater, the latter implying that mitochondrial cristae were more separated from each other. No significant differences in cristae alignment or cristae density were observed in the mitochondria of lactate-, succinate-, and butyrate-treated monocytes (Figure 3B). 

### 3.4. Lactate, Fumarate, Succinate, Butyrate, and Acetate Increase Δψ_m_ in Monocytes 

Once ultrastructural differences in mitochondrial cristae in response to metabolite stimulation were observed, we sought to evaluate whether the mitochondrial function of monocytes was also modified. Figure 4 shows a significant increment in Δψ_m_ from the first hour of metabolite stimulation, which was sustained at 4 h post metabolite stimulation, in all cases (Figure 4A). Of note, unstimulated monocytes comprise a mostly homogeneous population in terms of their Δψ_m_, whereas at later time points post-metabolite stimulation, cell subpopulations with different Δψ_m_, were observed (Figure 4B).

### 3.5. Fumarate Enhances Mitochondrial Respiration in Monocytes

Extracellular-flux analysis demonstrated changes in the use of energy-producing pathways upon metabolite treatment. Overnight pre-treatment of human monocytes with fumarate significantly increased mitochondrial ATP production rates (Figure 5) and at the same time decreased glycolytic ATP production rates (Figure 5), suggesting an increased capacity to use mitochondrial respiration for energy production. None of the other metabolites elicited notable changes upon treatment (Figure 5). 

## 4. Discussion

This study systematically analysed the response of mitochondria to the stimulation of monocytes, with an extracellular source of lactate (glycolysis), succinate, and fumarate (Krebs cycle), and butyrate and acetate (prototypical examples of microbiota-derived short-chain fatty acids), all of which harbour immunomodulatory properties [13,33,34]. The main finding is that each metabolite seems to induce a particular mitochondrial signature as defined by mitochondrial calcium influx, mitochondrial size and shape, cristae ultrastructure, Δψ_m_, and mitochondrial ATP production rate. Out of these findings, it is tempting to suggest that metabolites harbour information that is decoded by mitochondria, which in turn respond in the form of mitochondrial reprogramming. 

We have previously shown that monocyte exposure to fumarate induces mitochondrial Ca^2+^ uptake, induces mitochondrial fusion, increases Δψm, and induces mitochondrial cristae to become closer to each other, suggesting that mitochondrial activation might play a key role in fumarate-induced innate immune training [6,35].

Some metabolites act as signalling molecules [36,37,38], and there is increasing evidence that places mitochondria at the centre of the interaction between the immune system, metabolism, and the microbiota [21,39,40]. How metabolite-induced signalling is sensed by mitochondria and how these organelles respond to allow the flow of information between subcellular compartments, between cells, and between different biological domains, such as the interplay between the immune system and the microbiota, is an open question.

Lactate, succinate, fumarate, butyrate, and acetate are ligands of G-protein-coupled receptors (GPCR) [7,14,24,25] and GPCR-induced cell signalling involves cytoplasmic calcium influx [41,42,43]. We showed here that all the metabolites tested induced cytoplasmic calcium influx, suggesting GPCRs mediated cell signalling. It is worth noting that, whereas lactate-, succinate-, butyrate-, and acetate-induced calcium influx yielded similar levels of cytoplasmic calcium, fumarate induced comparatively higher levels, and individual differences in fumarate-induced cytoplasmic calcium influx amongst the different cell donors were more pronounced, as the higher standard deviations indicate (Figure 1A), suggesting individual differences in the expression of GRP109 or in GRP109-mediated signalling [7]. Nevertheless, the levels of intramitochondrial calcium that follow stimulation were similar for all metabolites tested, including fumarate (Figure 1B). 

Mitochondrial calcium influx/efflux regulates cells’ calcium signaling [44] as well as key Krebs cycle enzymes [26]. All metabolites tested induced mitochondrial calcium influx, and subtle differences in their kinetics responses were observed, perhaps reflecting metabolite-specific mitochondrial activity.

Mitochondrial dynamics is linked to mitochondrial and whole-cell homeostasis [45]; mitochondrial fission promotes mitochondrial biogenesis and allows damaged mitochondria to be excluded from the rest of the mitochondrial network, whereas mitochondrial fusion allows proteins and mtDNA exchange among mitochondria [29]. As previously shown, fumarate-exposure induces mitochondria to become more elongated, branched, and interconnected [35], which are all characteristics of mitochondrial fusion and enhanced oxidative phosphorylation [46,47,48]. Fumarate-exposure also increased both mitochondrial surface area, which may be an indicator of mitochondrial biogenesis [49], and closeness of mitochondrial cristae, which is tightly linked to mitochondrial bioenergetics, since mitochondria cristae contain the respiratory chain’s super complexes that can constantly change their arrangement and density, coordinating the kinetics of their biochemical reactions [50,51]. A model in which cristae within the same mitochondrion behave as independent bioenergetic units, preventing the failure of specific cristae form spreading dysfunction to the rest of the mitochondria, has been recently proposed [52]. In line with the mitochondrial changes observed in fumarate-treated monocytes, extracellular-flux-analysis experiments confirm that the use of oxidative phosphorylation for ATP production is enhanced by fumarate.

In contrast to fumarate, exposure to lactate-, succinate-, and butyrate induced monocytes’ mitochondria to become smaller and fragmented. These results seem to be in line with the previous finding, that succinate induces mitochondria fission in neurons and cardiomyocytes [53,54]. However, in the latter instances, mitochondrial fission takes place under conditions of ischemic stroke and cardiac ischemia, respectively, leading to the activation of cellular-stress programs [53,54]. In the present study, monocytes were exposed to succinate in the absence of any pathological stimuli and, thus, mitochondrial fission in monocytes may signal for a different outcome, such as monocyte/macrophage polarization [55]; butyrate-induced mitochondrial fission in monocytes appears to be in contrast to previous reports on mitochondrial fusion in hepatocytes [56], lymphoblastoid cell lines [57], and muscle cells [58]. Acetate stimulates mitochondrial biogenesis in adipocytes [59] and reduces oxidative phosphorylation in cardiomyocytes [60]; acetate as well as butyrate support β-cell metabolism and improve mitochondrial respiration, under oxidative stress [61]. Here, we showed that in monocytes exposed to acetate, mitochondrial cristae exhibit a wider separation, a mitochondrial ultrastructural feature associated with reduced oxidative phosphorylation [50,51]. These apparent discrepancies could be the result of differences in the experimental models (e.g., cell type, metabolites doses, pathological conditions, etc.). 

Finally, all the metabolites tested increased monocyte Δψ_m_ within the first hour post-treatment, in spite of differences in cristae ultrastructure. Of note, although Δψ_m_ increased in the monocyte population as a whole, monocyte subpopulations with different Δψ_m_ were observed, perhaps indicating heterogeneity in their response to extracellular metabolites; whether functional differences in monocytes are associated with particular Δψ_m_ remains to be analysed.

Taken together, this work outlines a possible mechanism of metabolite-dependent cell–cell communication, being a monocyte one of those cells, and between monocytes and microbiota (by means of the synthesis and export of short-chain fatty acids), through a process that we refer to as metabolite-induced mitochondrial reprogramming, in which host cells- and microbiota-derived metabolites engage G-protein-coupled receptors, leading to the influx of Ca^2+^ into the cytoplasm and into the mitochondria, increased Δψ_m_, induction of differential mitochondrial size, shape, cristae ultrastructure, and mitochondrial ATP production rates (summarized in Figure 6). However, in the case of acetate and butyrate, there is an alternative mechanism. SCFAs may enter the cell via the monocarboxylate transporter-1 (MCT-1), the sodium-coupled monocarboxylate transporter-1 (SMCT-1), and also by free diffusion through the cell membrane and the mitochondrial membranes; once inside, mitochondria SCFAs are used as substrates in mitochondrial β-oxidation and the citric acid cycle, being an important source of energy production [62,63,64]. MCTs transport SCFA in an H^+^-dependent manner, and the SMCT-1-mediated transport of SCFAs is coupled to a Na^+^/H^+^ exchanger, stimulating Cl^−^ and water absorption [65], thus, the observed changes in mitochondrial ultrastructure could also be the result of these ion movements [66].

While clear differences in mitochondrial morphology and function in response to different metabolites are observed, a limitation of this study is whether there is any correlation between metabolite-specific imprinting of mitochondrial activity and monocyte function, which still remains to be analysed.

## 5. Conclusions

The mitochondria from monocytes exposed in vitro to an external source of lactate, succinate, fumarate, acetate or butyrate differentially change their inner ultrastructure and activity, in a process that we refer to as metabolite-induced mitochondrial reprograming. This may have the potential to redirect cell activity.

## Figures and Tables

**Figure 1 biology-11-01132-f001:**
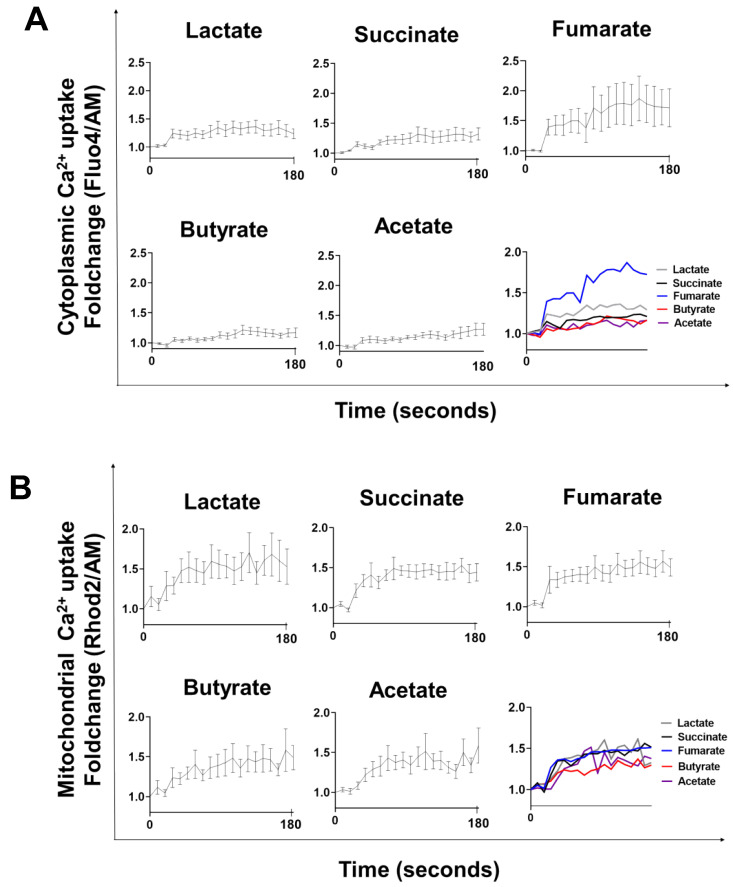
Lactate-, succinate-, fumarate-, butyrate-, and acetate-stimulation triggers cytoplasmic and mitochondrial calcium fluxes. Cytoplasmic (**A**) and mitochondrial (**B**) Ca^2+^ fluxes were assessed in macrophages by flow cytometry, using Fluo-4 and Rhod-2 molecular probes, respectively. Base levels of calcium were recorded for 30 s before metabolite exposure and then, in real time, for 180 s after metabolite exposure. Results are from eight independent experiments.

**Figure 2 biology-11-01132-f002:**
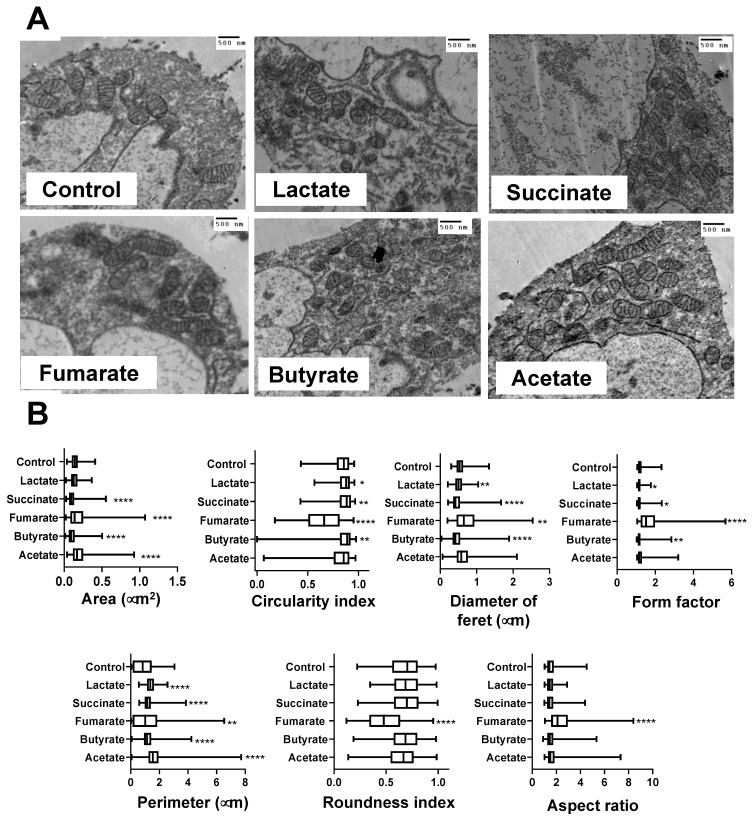
Mitochondria from lactate-, succinate-, and butyrate-stimulated monocytes were smaller and less complex than mitochondria from fumarate-stimulated cells. Monocytes stimulated with 100 μM of lactate, succinate, fumarate, butyrate, and acetate were processed for transmission electron microscopy (TEM). From TEM images (**A**), mitochondria dimensions were assessed by manually contouring mitochondrial structures (**B**). The mitochondria from at least 50 monocytes from three independent experiments were analysed. Statistical significance was analysed by ANOVA test and post hoc Tukey * *p* < 0.05, ** *p* < 0.01, **** *p* < 0.0001.

**Figure 3 biology-11-01132-f003:**
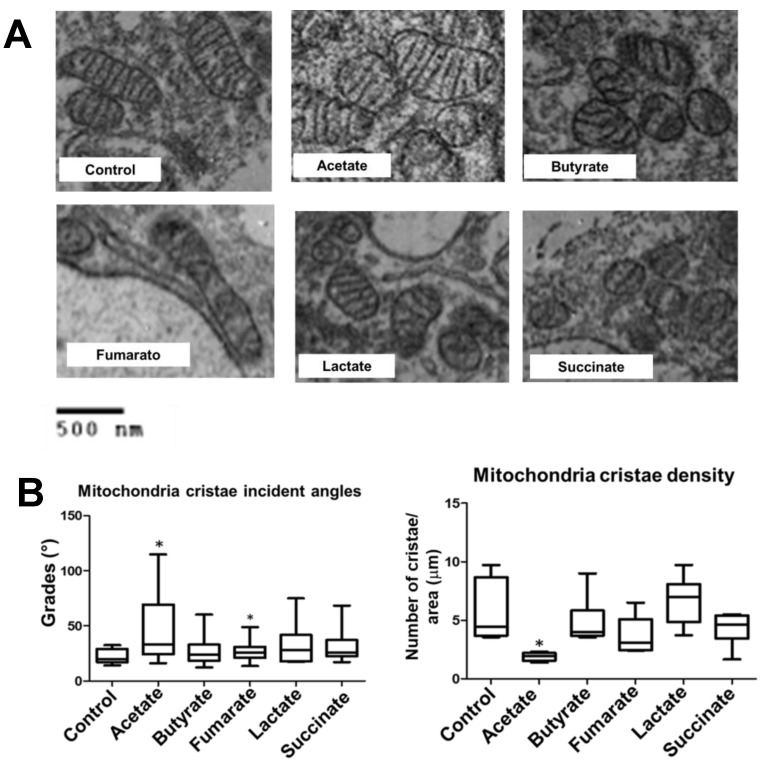
An intermediate metabolite from the Krebs cycle (fumarate) and a microbiota-derived short-chain fatty acid (acetate) induced opposite effects on the organisation of mitochondrial cristae. Monocytes were stimulated with lactate, succinate, fumarate, butyrate, or acetate for 3 h, fixed, and prepared for TEM. From TEM images, mitochondrial cristae ultrastructure was analysed. Over 600 mitochondria from at least 60 cells per condition from two independent experiments were analysed. Cristae density was calculated as the cristae number per surface area. Mean incident angle, as a function of mitochondria cristae closeness, was also analysed. (**A**) Representative TEM images of mitochondria cristae, (**B**) Cristae incident angles as a function of mitochondria cristae closeness; cristae density was calculated as the cristae number per surface area. Statistical significance was assessed by Kruskal–Wallis test and Dunn’s post hoc test * *p* < 0.05 vs. negative control. Statistical significance was assessed by Kruskal–Wallis test and Dunn’s post hoc test * *p* < 0.05 vs. negative control.

**Figure 4 biology-11-01132-f004:**
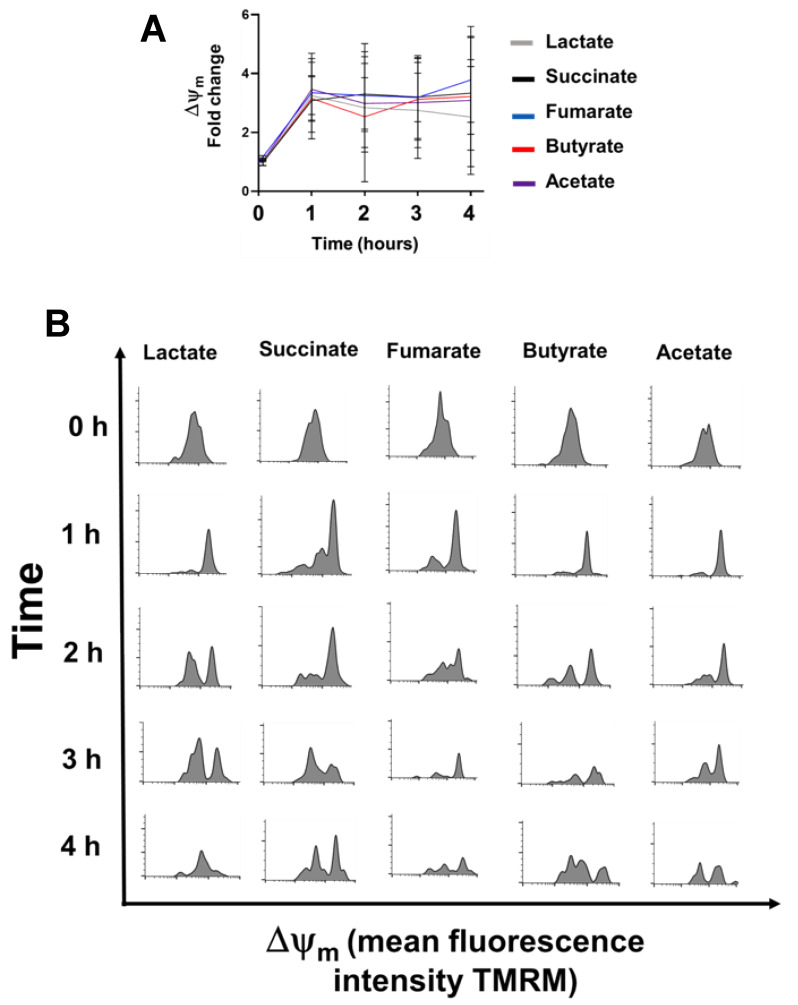
Lactate, succinate, fumarate, butyrate, and acetate induce mitochondria polarization in monocytes. Mitochondrial membrane potential (Δψ_m_) was assessed in human monocytes by flow cytometry, using the tetramethylrhodamine methyl ester molecular probe. For analysis, CD14+ cells were gated and TMRM mean fluorescence intensity (indicative of Δψ_m_) was recorded. (**A**) shows the Δψ_m_ fold change at 1, 2, 3, and 4 h post metabolite exposure, (**B**) shows representative histograms of TMRM fluorescence. Results are from eight independent experiments. Mitochondrial polarization was statistically significant (*p* < 0.001, Kruskal–Wallis and Dunn’s post hoc tests).

**Figure 5 biology-11-01132-f005:**
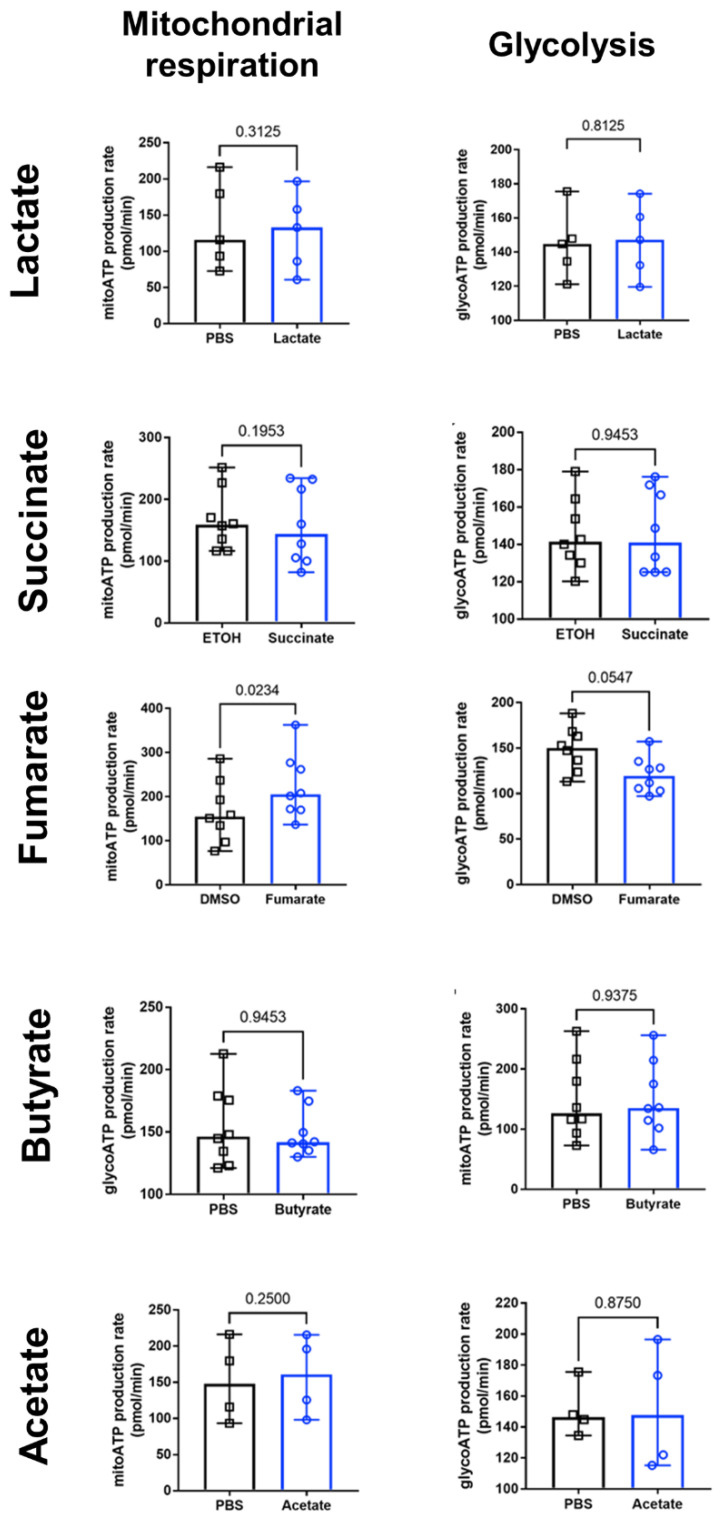
Increased mitochondrial ATP production rate in fumarate-treated human monocytes. Oxygen-consumption rates (OCR) and extracellular-acidification rates (ECAR) were measured on an XF96 extracellular-flux analyser upon sequential addition of oligomycin and rotenone/antimycin A to human monocytes pre-treated for 20 h with fumarate (*n* = 8), succinate (*n* = 8), butyrate (*n* = 8), lactate (*n* = 5), and acetate (*n* = 4). DMSO, ethanol, and PBS served as vehicle controls. Mitochondrial (mito) ATP production rate and glycolytic (glyco) ATP production rate was calculated from OCR and ECAR readings and plotted as bar graphs showing median and range. Data shown were obtained from four independent experiments. Wilcoxon paired signed rank test was performed between metabolite-treated sample and vehicle control, and a *p* value of <0.05 was considered significant.

**Figure 6 biology-11-01132-f006:**
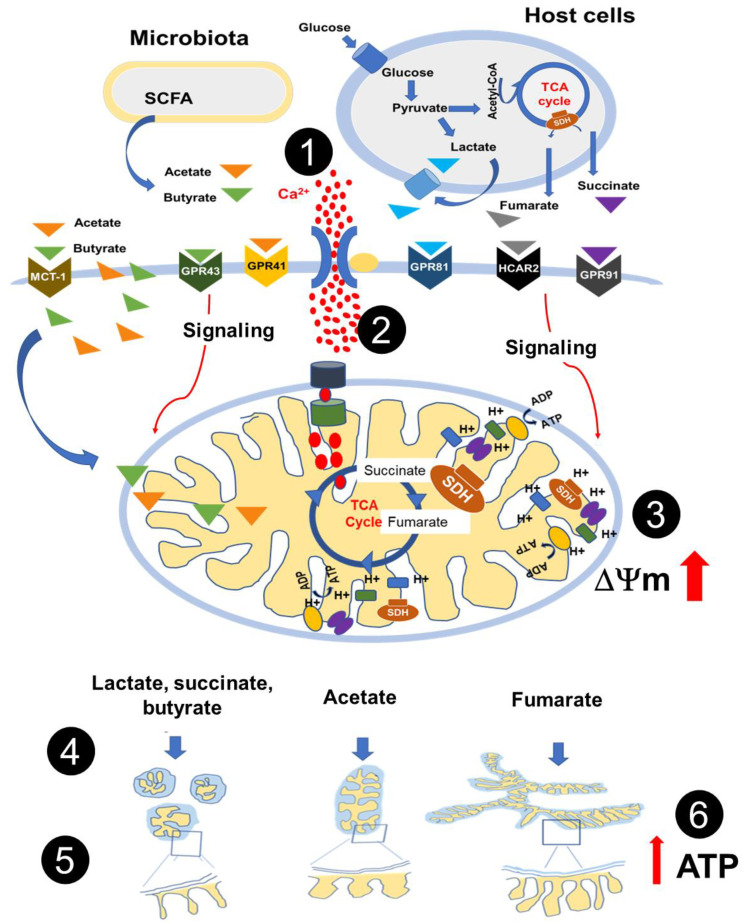
Metabolite-induced metabolic reprogramming of monocytes. An extracellular source of metabolites produced during glycolysis (lactate), during the TCA cycle (succinate and fumarate), and by the microbiota (butyrate and acetate) induced cytoplasmic Ca^2+^ influx (1) and mitochondrial calcium influx (2) within seconds, resulting in mitochondrial polarization (3), and differential mitochondrial shape and cristae ultrastructure that take place within hours (4 and 5); fumarate increases the production of mitochondrial ATP (6). In the particular case of acetate and butyrate, the mitochondrial ultrastructural changes could also be the result of MCT-1- and SMCT-1-mediated transport, coupled to Na^+^/H^+^ exchanger, and Cl^−^ and water absorption.

## Data Availability

The data presented in this study are available on request from the corresponding author.

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
