# Peer review of "Mitochondrial Ultrastructure and Activity Are Differentially Regulated by Glycolysis-, Krebs Cycle-, and Microbiota-Derived Metabolites in Monocytes"

_biology, 2022, doi:10.3390/biology11081132_

Round 1

Reviewer 1 Report

The present study analyzed the mitochondrial response to different intermediate metabolites of the Krebs cycle and glycolysis and microbiota-derived short-chain fatty acids. The authors evaluated the interaction of cellular and mitochondrial calcium flow, associating it with changes in the electrochemical potential of the mitochondrial membrane, as well as its morphology and capacity to supply ATP. The data are interesting and demonstrate the high capacity of mitochondrial interaction, signaling and remodeling for an effective regulation of cellular bioenergetics in human monocytes. However, some data need to be explored so that the proposed mechanism has more experimental basis.

Major points:

(1)   What were the criteria used to characterize healthy individuals? I strongly recommend that this information be better described in the material and methods.

(2)   For the proper determination of the source of ATP in the cell through measurements of the acidification rate (ECAR), it is necessary to isolate each of the possible sources of H+ using the equation described in the previously published article (PMDI:27031845). The authors use as cellular substrates: glutamine, glucose and pyruvate. In this situation, how to isolate the acidification from the pyruvate dehydrogenase (PDH) complex and glycolysis?

(3)   During the development of the experiments to evaluate the mitochondrial membrane pote             ntial using TMRM fluorescence, was any experimental control performed to validate whether the increase in fluorescence is directly related to the increase in membrane potential? For example, is this increase sensitive to the carbonyl cyanide 4-(trifluoromethoxy)phenylhydrazone (FCCP) uncoupler?

Minor points:

(1)   Review formatting error page 1 and line 40.

(2)   Revise and add to the Figures panels the nomenclatures: 2A, 2B, etc.

(3)   I recommend that authors review the manuscript and standardize which cell was used in each of the experiments: monocytes, macrophages or monocytes-derived macrophages?

(4)   I suggest that the authors add a justification to the definition of the concentration of metabolic intermediates used throughout the study.

Author Response

INSTITUTO POLITÉCNICO NACIONAL

ESCUELA NACIONAL DE CIENCIAS BIOLÓGICAS

Departamento de Inmunología

 June 30, 2022

Dear Biology MDPI Editors,

Please find enclosed the revised (R1) version of the manuscript “Mitochondrial ultrastructure and activity are differentially regulated by glycolysis-, Krebs cycle-, and microbiota-derived metabolites in monocytes” to be considered for publication in Biology MDPI.

We really appreciate the reviewer´s comments, we have addressed all those the best we could. We think that after this revision the manuscript has been improved and hope that the reviewers agree with this. 

The changes to the manuscript are as follow:

REVIEWER 1

The present study analyzed the mitochondrial response to different intermediate metabolites of the Krebs cycle and glycolysis and microbiota-derived short-chain fatty acids. The authors evaluated the interaction of cellular and mitochondrial calcium flow, associating it with changes in the electrochemical potential of the mitochondrial membrane, as well as its morphology and capacity to supply ATP. The data are interesting and demonstrate the high capacity of mitochondrial interaction, signaling and remodeling for an effective regulation of cellular bioenergetics in human monocytes. However, some data need to be explored so that the proposed mechanism has more experimental basis.

Major points:

  • What were the criteria used to characterize healthy individuals? I strongly recommend that this information be better described in the material and methods.

Healthy volunteers donating blood for research studies declared that they are not aware that they suffer from any persistent medical condition or infection which would affect their suitability to be a blood donor. At LSHTM they were also asked if they have been BCG vaccinated, and if they have lived abroad or visited a malaria-endemic area or ever had malaria.

This is now stated in the Materials and methods section

  • For the proper determination of the source of ATP in the cell through measurements of the acidification rate (ECAR), it is necessary to isolate each of the possible sources of H+ using the equation described in the previously published article (PMDI:27031845). The authors use as cellular substrates: glutamine, glucose and pyruvate. In this situation, how to isolate the acidification from the pyruvate dehydrogenase (PDH) complex and glycolysis?

Thank you very much for this comment. We have now added more detail about the inhibitors and procedures used to distinguish glycolytic and mitocondrial ATP production rates into the methods section for extracellular flux analysis.The calculations were based on the optimised protocol developed by Agilent and carried out by Wave software. We have added the following to the manuscript: “The addition of the ATP synthase inhibitor oligomycin inhibits mitochondrial ATP production thus reducing OCR and allowing the quantification of mitoATP production rate. ECAR levels measured in this assay combined with the buffer factor of the assay medium (determined by the manufacturer) were used to calculate total proton efflux rate (PER). Because mitochondrial respiration also contributes to ECAR levels, complex I and III of the electron transport chain were blocked by the addition of rotenone and antimycin A respectively. This combined with the PER was then used to calculate the glycoATP production rate.”

- During the development of the experiments to evaluate the mitochondrial membrane potential using TMRM fluorescence, was any experimental control performed to validate whether the increase in fluorescence is directly related to the increase in membrane potential? For example, is this increase sensitive to the carbonyl cyanide 4-(trifluoromethoxy)phenylhydrazone (FCCP) uncoupler?

Routinely, when performing mitochondrial membare potential and calcium influx assays, we use CCCP (Carbonyl cyanide 3-chlorophenylhydrazone) and Ionomycin, respectively, to make sure that the fluoresecent signal is indeed due to Dym and calcium influx. 

Thank you for pointing this out. We have now added  a brief paragraph explaining this in the materials and methos section, for both Dym and calcium infux assays.

Minor points:

Review formatting error page 1 and line 40.

Corrected

  • Revise and add to the Figures panels the nomenclatures: 2A, 2B, etc.

Corrected

  • I recommend that authors review the manuscript and standardize which cell was used in each of the experiments: monocytes, macrophages or monocytes-derived macrophages?

Thanks for pointing this out, in fact the cells used are monocytes. This has now been corrected in the whole text.

  • I suggest that the authors add a justification to the definition of the concentration of metabolic intermediates used throughout the study.

We appreciate your kind suggestion. The concentration of metabolites was based on relevant physiology  concentrations and on previous publications. This is now included within the text, along with some additional references.

Sincerely yours

  1. Javier Sánchez-García

Reviewer 2 Report

The Authors' results highlight the role of Short Chain Fatty Acids (SCFAs) on mitochondrial morphology and oxidative phosphorylation activity. Moreover, the Authors assert that SCFA induces mitochondrial reprogramming by metabolites bound to receptors couplet with G-protein. I would like to know if the Authors have considered the bulk SCFA movements across the energy-transducing membranes of mitochondria that produce changes in mitochondrial volume. Moreover, the uptake of SCFA, as a solute, can import cations, e.g. sodium, and modify the ion circuit of H+/Na+/Ca2++ through the membrane. Accordingly, the membrane potential is affected. I might suggest testing the effect of CGP-37157 an inhibitor of the mitochondrial Na+/Ca2+ exchanger.

Author Response

INSTITUTO POLITÉCNICO NACIONAL

ESCUELA NACIONAL DE CIENCIAS BIOLÓGICAS

Departamento de Inmunología

 June 30, 2022

Dear Biology MDPI Editors,

Please find enclosed the revised (R1) version of the manuscript “Mitochondrial ultrastructure and activity are differentially regulated by glycolysis-, Krebs cycle-, and microbiota-derived metabolites in monocytes” to be considered for publication in Biology MDPI.

We really appreciate the reviewer´s comments, we have addressed all those the best we could. We think that after this revision the manuscript has been improved and hope that the reviewers agree with this. 

The changes to the manuscript are as follow:

REVIEWER 2

Comments and Suggestions for Authors

The Authors' results highlight the role of Short Chain Fatty Acids (SCFAs) on mitochondrial morphology and oxidative phosphorylation activity. Moreover, the Authors assert that SCFA induces mitochondrial reprogramming by metabolites bound to receptors couplet with G-protein. I would like to know if the Authors have considered the bulk SCFA movements across the energy-transducing membranes of mitochondria that produce changes in mitochondrial volume. Moreover, the uptake of SCFA, as a solute, can import cations, e.g. sodium, and modify the ion circuit of H+/Na+/Ca2++ through the membrane. Accordingly, the membrane potential is affected. I might suggest testing the effect of CGP-37157 an inhibitor of the mitochondrial Na+/Ca2+ exchanger.

Thanks for your really interesting comment. Since all metabolites tested induced cytoplasmic calcium influx we considered  as very plausible that the differential changes in mitochondrial activity is due to the engagement of metabolte receptors.

Although very interesting, we think that testing mitochondrial inhibitors of Na+/Ca2+ exchangers is a bit out of the scope of the present manuscript but definetively we will keep your suggestion in mind for experiments in the near future.

We have now added a brief text in the discussion section to include the possibility of this alternative mechanism of SCFAs-induced mitochondrial activity ,along with a new reference.

Sincerely yours

F. Javier Sánchez-García

Reviewer 3 Report

The manuscript titled “Mitochondrial ultrastructure and activity are differentially regulated by glycolysis, Krebs cycle, and microbiota-derived metabolites in monocytes” by Perez-Hernandez et al. details an observational study looking at mitochondrial functional and ultrastructural effects in monocytes that are exposed to different metabolic fuels. They found some correlative effects pertaining to mitochondrial calcium handling, membrane potential, and ultrastructure. While the study does give us more information regarding monocytes and metabolites, the study lacks sufficient depth for me to endorse for publication. See my critique below for details.

The details regarding the metabolite addition to the cell cultures is missing. The methods state that RPMI media which contains glucose and amino acids capable of supporting both catabolic and anabolic metabolism. These constituents may confound the authors experimental results and is not discussed or controlled for, as far as I could tell.

Figure panels are not labeled with A, B, etc.

Why is the cytoplasmic Ca2+ signal noisier in the presence of fumarate compared to the other groups? Is the data more spread? Are population effects detectable? Including data points with each error bar will help answer this question.

Why are some of the EM images in Fig 3 and 4 so unfocused?

I only see two possible panels comprising Fig 4A and 4B but the text mentions 4C. Where is it?

In Fig 4, how is ECAR and OCR converted to ATP production rates?

Where there any whole-cell morphology changes associated with the metabolite supplementation?

With acetate, the authors show data indicating cristae density decreases. Is the matrix volume reciprocally increased? What is the matrix volume estimate for the other conditions?

Where the same effects seen with standard fumarate or succinate salts? Or were the esters required to deliver substrates from the media to the mitochondria? Do the ester breakdown products other than the substrate molecule lead to any changes in mitochondrial morphology or function?

What evidence do the authors have to support the idea that the mitochondrial calcium influx results are due to metabolite-specific mitochondrial signaling as stated on lines 301-305? Based on Fig 4A (membrane potential data), the increase in calcium is simply explained by the fact that the mitochondria are more polarized thus take up more calcium. Do the authors have data that show an increase in matrix Ca2+ with no change in membrane potential?

The connection between signaling and morphology/function is rather weak can be primarily explained via bioenergetic arguments. Do the authors have additional evidence to rule out the bioenergetic effects of these substrates to their monocyte cultures?

Minor

In the abstract on line 40, the Greek letters representing the membrane potential are missing.

Author Response

INSTITUTO POLITÉCNICO NACIONAL

ESCUELA NACIONAL DE CIENCIAS BIOLÓGICAS

Departamento de Inmunología

 June 30, 2022

Dear Biology MDPI Editors,

Please find enclosed the revised (R1) version of the manuscript “Mitochondrial ultrastructure and activity are differentially regulated by glycolysis-, Krebs cycle-, and microbiota-derived metabolites in monocytes” to be considered for publication in Biology MDPI.

We really appreciate the reviewer´s comments, we have addressed all those the best we could. We think that after this revision the manuscript has been improved and hope that the reviewers agree with this. 

The changes to the manuscript are as follow:

REVIEWER 3

Comments and Suggestions for Authors

The manuscript titled “Mitochondrial ultrastructure and activity are differentially regulated by glycolysis, Krebs cycle, and microbiota-derived metabolites in monocytes” by Perez-Hernandez et al. details an observational study looking at mitochondrial functional and ultrastructural effects in monocytes that are exposed to different metabolic fuels. They found some correlative effects pertaining to mitochondrial calcium handling, membrane potential, and ultrastructure. While the study does give us more information regarding monocytes and metabolites, the study lacks sufficient depth for me to endorse for publication. See my critique below for details.

The details regarding the metabolite addition to the cell cultures is missing. The methods state that RPMI media which contains glucose and amino acids capable of supporting both catabolic and anabolic metabolism. These constituents may confound the authors experimental results and is not discussed or controlled for, as far as I could tell.

Thanks for your comment. We have now tried to make this point clearer. Indeed, RPMI supplemented with glucose and aminoacids supports catabolic and anabolic metabolism. Our experimental approach is the addition of an exogenous source of lactate, fumarate, etc (at a final concentration of 100 mM). Regular RPMI (supplemented with glucose and glutamine, of course) without the addition of any external source of lactate, fumarate, etc was used as a negative control, i.e., the experiments were controlled for by culturing all cells in exactly the same culture media except for the referred metabolite, so that the results can be interpreted as a consequence of that single difference (the addition of an exogenous source of 100 mM of lactate, etc.). Some changes to the text have been introduced, in other to make this very important point clearer.

Figure panels are not labeled with A, B, etc.

We have now labeled the figures with A, B, etc. As appropriate.

Why is the cytoplasmic Ca2+ signal noisier in the presence of fumarate compared to the other groups? Is the data more spread? Are population effects detectable? Including data points with each error bar will help answer this question.

Thanks, yours is an interesting question. Each independent experiment was carried out on the same day, all the experimental conditions each time, with the same batch of reagents, the same flow cytometer etc. So, Ca2+ signal “noise” in the fumarate samples is likely to be due to fumrate itself. Wether there is a population effect is an interesting question (differential expression of fumarate receptors?, etc), beyod the scope of this manuscript. However, we will keep this question in mind for future research. We now briefly discuss this in the discussion section.

Why are some of the EM images in Fig 3 and 4 so unfocused?

Thanks for pointig this. We have now susbstituted some EM images with appropriate focussing. (Included as separate files). Besides, figures have been rearranged so that the sequences are the same for panel A and B.

I only see two possible panels comprising Fig 4A and 4B but the text mentions 4C. Where is it?

You are right, our mistake. There are only 4A and 4B (and without proper labeling). This has now been corrected in the figure, in the figure legend and in the text.

In Fig 4, how is ECAR and OCR converted to ATP production rates?

The calculations were based on the optimised protocol developed by Agilent and carried out by Wave software. We have added the following to the manuscript: “The addition of the ATP synthase inhibitor oligomycin inhibits mitochondrial ATP production thus reducing OCR and allowing the quantification of mitoATP production rate. ECAR levels measured in this assay combined with the buffer factor of the assay medium (determined by the manufacturer) were used to calculate total proton efflux rate (PER). Because mitochondrial respiration also contributes to ECAR levels, complex I and III of the electron transport chain were blocked by the addition of rotenone and antimycin A respectively. This combined with the PER was then used to calculate the glycoATP production rate.”

Where there any whole-cell morphology changes associated with the metabolite supplementation?

We concentrate our efforts on mitochondrial morphology. However, as far as we can tell, no evident differences in whole-cell morphology were observed.

With acetate, the authors show data indicating cristae density decreases. Is the matrix volume reciprocally increased? What is the matrix volume estimate for the other conditions?

We didn´t assessed matrix volume but mitochondrial area. Mitochondrial area increased upon acetate treatment (fig. 2), which would explain, at least in part, the decrease in cristae density. Of note, lactate treatment also significantly incresed mitochondrial area, without a significant decrease in cristae density and thus cristae density changes are not solely the result of changes in mitochondrial area. This was not previously discused.

Thanks for this insightful observation. We have now added a few lines in the discussion section.

Where the same effects seen with standard fumarate or succinate salts? Or were the esters required to deliver substrates from the media to the mitochondria? Do the ester breakdown products other than the substrate molecule lead to any changes in mitochondrial morphology or function?

Only mono-methyl fumarate and diethyl succinate were used in the present study and not standard salts. These molecules have previously been used when analysing the effect of fumarate and succinate on cells, as in references 10 and 56, respectively.

What evidence do the authors have to support the idea that the mitochondrial calcium influx results are due to metabolite-specific mitochondrial signaling as stated on lines 301-305? Based on Fig 4A (membrane potential data), the increase in calcium is simply explained by the fact that the mitochondria are more polarized thus take up more calcium. Do the authors have data that show an increase in matrix Ca2+ with no change in membrane potential?

This paragraph (lines 301-305) is based on the curren knowledge that mitochondrial calcium influx/efflux regulates cells calcium signaling (reference 45) and also that mitochondrial calcium regulates the activity of key Krebs cycle enzymes (reference 26). Since mitochondrial calcium influx takes place upon metabolite stimulation it follows that regulation of calcium signaling and regulation of the activity of Krebs cycle enzymes is also taking place. 

This paragraph also mention our finding that subtle differences in the calcium influx kinetics upon exposure to the different metabolites tested were observed.

Perhaps the last part of the paragraph: “perhaps reflecting metabolite-specific mitochondrial signalling” is a bit of an overstatement, and we thank you for pointing this out.

Accordingly, we have now changed this last part for “perhaps reflecting metabolite-specific mitochondrial activity”.

No, we do not have any data showing an increase in matrix Ca2+ with no change in membrane potential.

The connection between signaling and morphology/function is rather weak can be primarily explained via bioenergetic arguments. Do the authors have additional evidence to rule out the bioenergetic effects of these substrates to their monocyte cultures?

In line with the previous point, we have now substituted mitochondrial signaling for mitochondrial activity leaving open the posibility for signaling or bionergetic arguments.

Minor

In the abstract on line 40, the Greek letters representing the membrane potential are missing.

This has now been corrected

Sincerely yours

F. Javier Sánchez-García

Round 2

Reviewer 1 Report

The authors revised the manuscript and added important information for a better understanding of the reader and greater solidity of the data.

Author Response

REVIEWER 1

The authors revised the manuscript and added important information for a better understanding of the reader and greater solidity of the data.

Thanks a lot for your comments.

Reviewer 2 Report

Authors assert that "We have now added a brief text in the discussion section to include the possibility of this alternative mechanism of SCFAs-induced mitochondrial activity ,along with a new reference." In my opinion, the sentence of Authors in the discussionit is not exhaustive for my requests on the bulk SCFA movements across the energy-transducing membranes of mitochondria that produce changes in mitochondrial volume as a result of changes in the ion circuit of H+/Na+/Ca2++ through the inner mitochondrial membrane.

Author Response

REVIEWER 2

Comments and Suggestions for Authors

Authors assert that "We have now added a brief text in the discussion section to include the possibility of this alternative mechanism of SCFAs-induced mitochondrial activity, along with a new reference." In my opinion, the sentence of Authors in the discussion it is not exhaustive for my requests on the bulk SCFA movements across the energy-transducing membranes of mitochondria that produce changes in mitochondrial volume as a result of changes in the ion circuit of H+/Na+/Ca2++ through the inner mitochondrial membrane.

This is a mechanism that we had not considered at all, and it is a really interesting one, that will require further study.  Thanks for pointing this out.

We have now attempted to provide an alternative explanation to our results regarding acetate and butyrate, without being exhaustive, since the subject is a whole area of research on its own.

We included a new paragraph that reads: “However, in the case of acetate and butyrate, there is an alternative mechanism. SCFAs may enter the cell via the monocarboxylate transporter-1 (MCT-1), the sodium-coupled monocarboxylate transporter-1 (SMCT-1) and also by free diffusion through the cell membrane and the mitochondrial membranes, once inside mitochondria SCFAs are used as substrates in mitochondrial β-oxidation and the citric acid cycle, being an important source of energy production (63-65). MCTs transport SCFA in an H+-dependent manner, and the SMCT-1-mediated transport of SCFAs is coupled to a Na+/H+ exchanger, stimulating Cl− and water absorption (66) thus, the observed changes in mitochondrial ultrastructure could also be the result of these ion movements (67).” (lines 384-393).

We Added 5 new references.

Figure 6 as well as figure 6 legend were changed accordingly.

We consider that although we are unable right now to do the experiment you suggested, providing an alternative explanation within the discussion section, adds value to our manuscript, and we thank you for that.

Reviewer 3 Report

The authors addressed most of my concerns. That said, a few issues still remain to be addressed.

Many of the new EM images are appear faded with poorer contrast than the originals. I advise including to sharper images. On this note, the new butyrate panel image in Fig 3 shows nothing.

About fumarate and cytosolic Ca2+ levels, the response letter states, "We now briefly discuss this in the discussion section." However, I didn't see any such discussion. I only saw discussion topics related to fumarate that were already in the unrevised manuscript that I reviewed. If I missed the relevant text, what lines are they at?

Author Response

REVIEWER 3

The authors addressed most of my concerns. That said, a few issues still remain to be addressed.

Many of the new EM images are appear faded with poorer contrast than the originals. I advise including to sharper images. On this note, the new butyrate panel image in Fig 3 shows nothing.

Thanks. This is a very important issue since mitochondrial ultrastructure is a key component of the present manuscript. We have gone to review all our original EM images and hope we have come up with a better version of both figure 2 and figure 3. The new version looked fine to us. Not sure if figures lost resolution when copied within the text. This time we have corroborated that butyrate panel in figure 3 actually shows something.

About fumarate and cytosolic Ca2+ levels, the response letter states, "We now briefly discuss this in the discussion section." However, I didn't see any such discussion. I only saw discussion topics related to fumarate that were already in the unrevised manuscript that I reviewed. If I missed the relevant text, what lines are they at?

I apologise, during R1 edition I failed to add that paragraph. It is now included (lines 324-331) and it reads as follow:

“It is worth noting that, whereas lactate-, succinate-, butyrate- and acetate-induced calcium influx yielded similar levels of cytoplasmic calcium, fumarate induced comparatively higher levels, besides, individual differences in fumarate-induced cytoplasmic calcium influx amongst the different cell donors were more pronounced, as the higher standard deviations indicate (Fig. 1A), suggesting individual differences in the expression of GRP109 or in GRP109-mediated signalling (7). Nevertheless, the levels of intramitochondrial calcium that follow stimulation were similar for all metabolites tested, including fumarate (Fig. 1B).”
